# Coffee Consumption and the Risk of Metabolic Syndrome in the ‘Seguimiento Universidad de Navarra’ Project

**DOI:** 10.3390/antiox12030686

**Published:** 2023-03-10

**Authors:** María J. Corbi-Cobo-Losey, Miguel Á. Martinez-Gonzalez, Anne K. Gribble, Alejandro Fernandez-Montero, Adela M. Navarro, Ligia J. Domínguez, Maira Bes-Rastrollo, Estefanía Toledo

**Affiliations:** 1Department of Preventive Medicine and Public Health, University of Navarra, Irunlarrea 1, 31008 Pamplona, Spain; 2CIBER Fisiopatología de la Obesidad y Nutrición, Av. Monforte de Lemos, 3-5. Pabellón 11. Planta 0, 28029 Madrid, Spain; 3IdiSNA, Navarra Institute for Health Research, Irunlarrea 3, 31008 Pamplona, Spain; 4Department of Nutrition, Harvard T. H. Chan School of Public Health, 677 Huntington Ave, Boston, MA 02115, USA; 5Sydney Medical School, Edward Ford Building (A27), University of Sydney, Fisher Rd., Sydney, NSW 2006, Australia; 6Department of Occupational Medicine, University of Navarra Clinic, Av. Pio XII, 36, 31008 Pamplona, Spain; 7Department of Cardiology, Complejo Hospitalario de Navarra, Servicio Navarro de Salud Osasunbidea, Irunlarrea 3, 31008 Pamplona, Spain; 8Geriatric Unit, Department of Internal Medicine and Geriatrics, University of Palermo, 90134 Palermo, Italy

**Keywords:** metabolic syndrome, coffee, prospective cohort, SUN Project

## Abstract

(1) Background: Metabolic Syndrome (MetS) affects over a third of the United States population, and has similar prevalence in Europe. Dietary approaches to prevention are important. Coffee consumption has been inversely associated with mortality and chronic disease; however, its relation to the risk of MetS is unclear. We aimed to investigate the association between coffee consumption and incident MetS in the ‘Seguimiento Universidad de Navarra’ cohort. (2) Methods: From the SUN project, we included 10,253 participants initially free of MetS. Coffee consumption was assessed at baseline, and the development of MetS was assessed after 6 years of follow-up. All data were self-reported by participants. MetS was defined according to the Harmonizing Definition. We used multivariable logistic regression models to estimate odds ratios and 95% confidence intervals for incident MetS according to four categories of coffee consumption: <1 cup/month; ≥1 cup/month to <1 cup/day; ≥1 cup/day to <4 cups/day; ≥4 cups/day. (3) Results: 398 participants developed MetS. Coffee consumption of ≥1 to <4 cups/day was associated with significantly lower odds of developing MetS (multivariable adjusted OR = 0.71, 95% CI (0.50–0.99)) as compared to consumption of <1 cup/month. (4) Conclusions: In a Mediterranean cohort, moderate coffee consumption may be associated with a lower risk of MetS.

## 1. Introduction

Metabolic syndrome (MetS) is a confluence of cardiovascular risk factors incorporating insulin resistance, dyslipidaemia, obesity, and hypertension. MetS is estimated to affect approximately one quarter of the global population [1] despite definitional discrepancies. The most widely accepted definition is the Harmonizing Definition provided by the International Diabetes Federation (IDF) Task Force on Epidemiology and Prevention and the American Heart Association/National Heart, Lung, and Blood Institute (AHA/NHLBI) [1]. Three out of five MetS criteria must be met in order to make a diagnosis. In the United States, MetS prevalence increased by almost 30% in just over 30 years [2]. European studies have described the prevalence to be 23.9% in men and 24.6% in women [3]. As for Spain, the prevalence of MetS was estimated at 22.7% based on 2008–2010 data using the Harmonized Definition [4]. This high prevalence is troubling, as the presence of MetS is associated with an increased risk of new onset cardiovascular disease (CVD) and all-cause mortality [5]. The presence of MetS doubles the risk of CVD over 5–10 years and increases the risk of type 2 diabetes by fivefold or more over a lifetime [1]. It is therefore of clinical relevance to work towards its prevention [1].

Diet and lifestyle changes are an important tool in the prevention of chronic disease. The Mediterranean diet is a pattern of food consumption that has proven its potential to protect cardiovascular health [6] as well as protect against MetS [7]. It differs from other dietary patterns because of its comparatively higher consumption of fats (mainly olive oil) and moderate wine intake [8].

Coffee is one of the most widely consumed beverages around the world and in Europe it is consumed at a rate of approximately 8.3 kg per year per capita [9]. Spain ranks 23rd among countries with the highest per capita coffee consumption [10]. Coffee contains a plethora of micronutrients that have been associated with cardiometabolic health [11], and its consumption causes short-term physiological changes which also affect the cardiovascular system. Coffee consumption and coffee micronutrient components have been suggested to improve glucose metabolism, lower inflammation, and decrease liver damage in animal models [12,13]. Further evidence suggests that it improves endothelial function, aids in loss of fat mass and is associated with favorable plasma biomarkers of metabolic and inflammatory pathways [14,15,16]. A recent metanalysis of randomized controlled trials by Ramli et al. investigated the effects of coffee consumption on anthropometric measurements, glycemic indices, lipid profiles, and blood pressure and found all of these parameters were improved by green coffee extract supplementation [17]. It is of great interest to continue to investigate the relationship between coffee consumption and the development of MetS, especially given that there is little prospective epidemiological evidence on the association between coffee consumption and MetS incidence [18].

There are only three published prospective cohort studies on the relationship between coffee and MetS [19,20,21]. Two of these did not find any significant association between coffee consumption and MetS [19,21]. The other pointed towards a protective effect of coffee consumption on MetS, but its results were not statistically significant [20]. These previous studies were heterogenous in several aspects, including the average baseline age (13–60 years), the average follow-up time (9–23 years), the way that they assessed coffee consumption, and the adjustment for important confounders such as dietary factors and physical activity levels. None of the mentioned studies adjusted for the overall dietary pattern, which may have introduced substantial confounding as coffee drinking habits have been associated with other dietary habits [22].

There are two main varieties of coffee: Arabica and Robusta. The former is more acidic and is predominantly produced in Latin America [23]; the latter is stronger but less acidic and is mainly found in West Africa and Southeast Asia and used in soluble coffee production [24]. During the roasting process, the Maillard reaction leads to the oxidative polymerization and degradation of phenolic compounds [25]. This same process reduces the quantity of carbohydrates, proteins, chlorogenic acid, and free amino acids but increases the lipid and mineral content as well as the caffeine and trigonelline content [26]. The method of coffee preparation has its own role in determining the concentrations of bioactive substances such as diterpenes [27], caffeine, or polyphenols [28]. For example, Moeenfard et al. found that boiled coffee showed the highest diterpene esters concentration, whereas filtered and instant brews showed the lowest concentrations [27].

Our aim was to address the association between coffee consumption and MetS in a prospective cohort of middle-aged Spanish adults and to evaluate the role of the overall dietary pattern in this association.

## 2. Materials and Methods

### 2.1. Study Population

The sample for this study was taken from the SUN project. SUN is a cohort of university graduates with open enrollment since December 1999. Participants complete an extensive questionnaire when joining the study and are asked to complete follow-up questionnaires every two years thereafter. Further details on the design of the SUN cohort can be found elsewhere [29]. At the time of data analysis for our study, a total of 22,894 participants had been recruited to the cohort. Participants were excluded for various reasons (Figure 1): 990 who had been followed for less than the minimum of 6 years for possible outcome assessment; 165 who had died prior to outcome assessment; 5137 who had not completed the 6- or 8-year follow-up questionnaire; 4312 who already had at least one component of MetS at enrollment; 1130 who had an energy intake outside of the predefined limits for realistic energy intake [30]; 627 who left more than 15 items blank in the food-frequency questionnaire; 68 who had previous cardiovascular disease; 211 who had previously suffered from cancer, and, 1 whose increased waist circumference during pregnancy confounded the assessment of MetS criteria. The remaining 10,253 participants were included in the study.

### 2.2. Coffee Consumption

A semi-quantitative food-frequency questionnaire (FFQ) was used to assess diet [31,32,33]. This previously validated questionnaire included questions on both caffeinated and decaffeinated coffee (cup size 50 cm^3^). Participants were asked to estimate the average number of cups consumed according to nine categories: ‘never/seldom’, ‘1–3 per month’, ‘1 per week’, ‘2–4 per week’, ‘5–6 per week’, ‘1 per day’, ‘2–3 per day’, ‘4–6 per day’, and ‘6+ per day’. We categorized coffee consumption into four variables (<1 cup/month, ≥1 cup/month to <1 cup/day, ≥1 cup/day to <4 cups/day, ≥4 cups/day).

### 2.3. Metabolic Syndrome

Data related to the components of metabolic syndrome were assessed on the third and fourth questionnaires, distributed approximately six and eight years after entry into the cohort. Participants self-reported recent measurements of waist circumference, blood pressure, HDL and LDL cholesterol, and fasting glucose, as well as diagnoses of hypertension and diabetes and any medicines they were using. MetS was defined according to the Harmonizing Definition from the IDF and AHA/NHLBI. That is, it was defined as the presence of three of the following five criteria [1]: (a) central adiposity (waist circumference ≥94 cm for men and ≥80 cm for women, as defined for the European population); (b) elevated triglycerides (≥150 mg/dL or undergoing treatment for hypertriglyceridemia); (c) reduced HDL cholesterol (<40mg/dL for men and <50 mg/dL for women or undergoing treatment for reduced HDL cholesterol); (d) high blood pressure (systolic ≥130 mmHg, diastolic ≥85 mmHg, or undergoing pharmacological treatment for hypertension); and (e) elevated fasting glucose (≥100 mg/dL or undergoing pharmacological treatment for hyperglycemia). Participants were provided with a measuring tape and a set of instructions on how to measure their waist circumference [34]. Each component of MetS was addressed as a dichotomous yes/no variable. The resulting ascertainment of MetS has been previously validated in our cohort [35]. MetS was primarily assessed based on data from the 6-year follow-up questionnaire. In the case of missing information, we used the 8-year follow-up questionnaire to replace missing data. 

### 2.4. Other Covariates

The SUN project uses a thorough baseline questionnaire to assess all of the individual characteristics of the participants. These include socio-demographic factors such as age, sex, and years at university; health–lifestyle factors such as smoking habits, siesta habits and hours of TV per day; and clinical history, including previous diagnoses of depression and family history of diabetes. Anthropometric variables of height and weight are assessed from which the body mass index (BMI) is then calculated. The average daily energy intake (kcal) and average daily alcohol intake (g/day) were derived from the FFQ based on Spanish food composition tables [36,37]. Important to our study, the baseline questionnaire included a 17-item questionnaire on physical activity, from which total leisure time physical activity (METs–h/week) was derived [38]. The questionnaire also asked whether the participants added sugar to beverages. Furthermore, the FFQ allowed for adherence to the Mediterranean diet to be assessed in accordance with the Mediterranean diet scale proposed by Trichopoulou et al. [39], which was modified to exclude the alcohol component (range: 0–8 points). Participants received one point for consuming more than the sex-specific median in each of the six traits typical of the traditional Mediterranean diet: vegetables, legumes, fruit and nuts, whole grain cereals, fish, and ratio of monounsaturated:saturated fats. One point was assigned for consuming less than the median in either of the two components considered contrary to the traditional Mediterranean diet: meat and dairy products. Coffee consumption is not included in this Mediterranean diet score.

### 2.5. Statistical Analysis

To address the association between coffee consumption and the risk of MetS, we used non-conditional logistical regression models. We fitted a crude model without adjusting for covariables and then another model adjusted for age and sex. Subsequently, we fitted an additional model adjusting for all other covariates: energy intake (continuous), adherence to a Mediterranean diet (continuous), alcohol intake (three categories), BMI (continuous), physical activity (quartiles), hours of watching TV (continuous), smoking (three categories), pack-years of smoking (continuous), previous depression (yes/no), years of university (continuous), hours of siesta (over or under half an hour), added sugar in drinks (yes/no), and family history of diabetes (yes/no). We addressed the goodness of fit (calibration) of the final model with the Hosmer–Lemeshow test [40].

We assessed the interactions between coffee consumption and sex, BMI (<25 kg/m^2^/≥25 kg/m^2^), age (continuous) and adherence to the Mediterranean diet (continuous) with the likelihood ratio test.

We repeated our analyses separately for caffeinated and decaffeinated coffee consumption with the aforementioned adjustments. In addition, the results for caffeinated coffee consumption were adjusted for decaffeinated coffee consumption and vice versa.

We deemed a *p* value below 0.05 to be statistically significant.

## 3. Results

Our analyses included 10,253 participants. Of these, 398 developed MetS. The baseline characteristics of participants according to the categories of coffee consumption are described in Table 1. The participants with higher coffee consumption were older and also included a moderately greater percentage of smokers, a moderately higher average BMI, and a higher total energy intake, as well as a better score of adherence to the Mediterranean diet. Participants who consumed ≥4 cups of coffee a day were more likely to add sugar to their beverages compared to those who consumed <1 cup per month.

Table 2 shows the odds ratios (OR) (95% confidence interval) for the risk of developing MetS by categories of coffee consumption. In comparison with the control category (<1 cup of coffee per month), participants who consumed one to less than four cups of coffee per day showed significantly lower odds of developing MetS (OR ≥ 1 cup/d to <4 cups/d 0.71, 95% CI (0.50–0.99)). The *p* for trend was not statistically significant, nor was the continuous association between one more cup of coffee per day and the risk of MetS.

We observed no significant interactions of coffee consumption with age, sex, BMI, or adherence to the Mediterranean diet.

When we separated caffeinated coffee consumption from decaffeinated coffee consumption, the point estimates across categories of caffeinated and decaffeinated coffee consumption were similar to what had been observed for total coffee consumption; however, the associations were slightly attenuated, and they were no longer significant. These results are shown in Table 3 for caffeinated beverages and Table 4 for non-caffeinated beverages.

## 4. Discussion

Our results suggest that moderate coffee consumption (≥1 cup/d to <4 cups/d) may be associated with a decreased risk of developing MetS in a multivariable model. We did not find any remarkable difference between caffeinated and decaffeinated coffee consumption. Although no single category yielded statistically significant results, there was a trend towards an overall protective effect once covariates were accounted for.

Previously published prospective studies on the association between coffee consumption and the development of MetS did not report significant results. A comparison of the results is difficult due to a high degree of heterogeneity: not only did they use different definitions for MetS, but they also used a vastly different categorization of coffee consumption. The Tromsø study [21], similar to our study, categorized coffee consumption in categories of cups/day; however, the ARIC study [20] grouped coffee consumption by quintiles, and the Amsterdam study [19] did not give details about their categorization of average long-term coffee consumption. It is important to note that our study was the only study to suggest an association that remained statistically significant after multivariate adjustment and was also the study that adjusted for the most covariables. Most importantly, we were able to limit confounding by adjusting for physical activity levels and overall dietary pattern.

A fundamental difference between the previous studies was their use of differing as well as modified definitions of MetS, leading to altered inclusion criteria and eventual inconsistencies in the data. The Tromsø and Amsterdam studies used modified versions of the NCEP ATP III definition [41], whereas the ARIC study used the American Heart Association guidelines [42]. 

Furthermore, the previous studies looked at data from the United States, the Netherlands, and Norway, whereas our cohort is from Spain. This spread of geographical locations is relevant for our study question as coffee consumption patterns - including the type of coffee bean and the method of preparation - may be different in the United States and Northern Europe as compared to Spain. For example, filtered, boiled, and instant coffee brewing methods predominate in Norway [43,44], whereas espresso and percolator coffee predominate in Spain [45]. It has been found that different brewing methods yield different results in terms of the extraction of antioxidants, caffeine, and micronutrients [46,47,48]. The preparation method and the type of coffee may impact the different compounds found within coffee. For example, diterpenes are largely removed during the filtering process [27] and are more present in the arabica bean variety [49]. The roasting process and the method of preparation may also be important for disease prevention, but we were not able to differentiate between roasting or brewing methods. 

In addition, the average baseline age differed greatly between the study samples of the previous studies. The ARIC study participants were middle-aged (mean age 53.6 years), whereas those included in the Amsterdam Study were much younger (27 years old at baseline). The median age of participants in the Trømso study was 30–39 years, so this latter study more closely resembles our own contribution. Inconsistent results across different age groups may suggest a critical time window for coffee consumption in MetS prevention.

There are several mechanisms that may explain the potential protective effect of coffee consumption on MetS incidence. Primarily, coffee consumption has been inversely associated with chronic inflammation, which is an underlying condition of MetS [16,50]. A significant association has been found between caffeine and reduced inflammation in animal studies [51]. Moreover, specific components found in coffee have been linked to beneficial biological processes. So, melanoidins have been shown to have both antioxidant and anti-inflammatory properties [52], chlorogenic acids have been shown to increase the bioavailability of NO and inhibit the activity of the angiotensin-converting enzyme [53], and diterpenes, especially cafestol and kahweol, have also been associated with the downregulation of inflammatory mediators [54].

Interestingly, coffee has previously been associated with hypertension, one of the MetS components, due to caffeine’s ability to increase blood pressure [55]. However, this effect was found for much higher concentrations of caffeine than those found in regular coffee consumption, which is also affected by extraction methods and methods of administration [56]. There is additional controversy as caffeine could actually alleviate hypertension by binding to the A1 adenosine receptors [57]. Meanwhile other studies have shown a quick adaptative response against the pressor effect as well as a tolerance among regular consumers of coffee [58]. In any case, coffee contains compounds besides caffeine. For example, it is rich in anti-hypertensive minerals (vitamin E, niacin, potassium, and magnesium) [59]. These other compounds could counteract the potential hypertensive effect of caffeine and could even be responsible for a beneficial effect of coffee consumption in terms of hypertension prevention. 

Coffee consumption has previously been linked to reduced risk of type 2 diabetes [60] and improved glucose metabolism as it reduces the area under the glucose curve and increases the insulin response [61]. High coffee consumption has been associated with increased insulin secretion, insulin sensitivity, and β-cell function [62], in part due to caffeine [61] and in part due to coffee’s effects on adiponectin [63]. As trigonelline has been found to reduce glucose and insulin levels at higher concentrations than those normally found in coffee, it is coherent to think that the long-term trigonelline intake from coffee may also exert a beneficial role [64].

Coffee consumption has also been associated with a lower risk of developing obesity [65] and with favorable changes in BMI and waist circumference over time [66,67]. It is possible that coffee may contribute to weight control through its capacity to reduce fatigue. This could lead to an increase in total energy expenditure. Caffeine has been observed to increase energy expenditure and therefore lead to weight loss by increasing thermogenesis [68]. The polyphenol content could be responsible for the reduced risk of dietary-induced obesity by increasing energy expenditure through their regulation of the expression of lipogenic enzymes [69].

The strengths of our study include its prospective design and our ability to account for a large range of potential confounders in our analyses. Nevertheless, we acknowledge that our results did not show any evidence of a linear trend and were not consistent with the previous literature. Dose–response trends and consistency are usually included among the causality criteria in observational studies, and they were not met in this study. Therefore, our findings can only be considered as suggestive of a potential protection, and need to be replicated in independent cohorts with good control of confounding.

We recognize several limitations of our study. First, coffee consumption was self-reported rather than quantified by objective means. This method of data collection was necessary due to the large size of the SUN cohort. It may be preferable to the time-limited methods of direct observation or food diary completion because it allows participants to report ‘usual consumption over the last year’. Importantly, the FFQ used by the SUN Project has previously been validated [31,32,33]. A second limitation is that participants’ coffee consumption was only reported at the beginning of the study, and we were not able to account for possible changes over time. Nevertheless, it has been suggested that the coffee consumption habit remains relatively constant in adulthood [70]. Third, we did not differentiate between coffee bean varieties or the processing or brewing methods. In Spain, the most frequent type of preparation has been reported to be unfiltered coffee (espresso or percolator) [45]. Fourth, information for the assessment of MetS criteria was self-reported, though this too has been previously validated in our cohort [35]. Fifth, it could be argued that the incidence of MetS that we observed (4.15%) was lower than the expected incidence among the general population. This reduction is likely because of the selection process (excluding all baseline criteria of MetS) as well as the low BMI, younger age, and higher educational level of our cohorts’ participants. In this way, our sample was not representative of the general population. Finally, due to the observational nature of our study, confounding cannot completely be ruled out even though we adjusted our results for a wide range of confounders.

## 5. Conclusions

In conclusion, moderate regular coffee consumption might reduce the risk of developing MetS, but further evidence is needed. There are physiological mechanisms that support a relationship between coffee and reduced risk of MetS. However, the currently available cohort studies are scarce. Our study is the first to find some statistically significant associations, though further prospective studies are warranted to confirm these results.

## Figures and Tables

**Figure 1 antioxidants-12-00686-f001:**
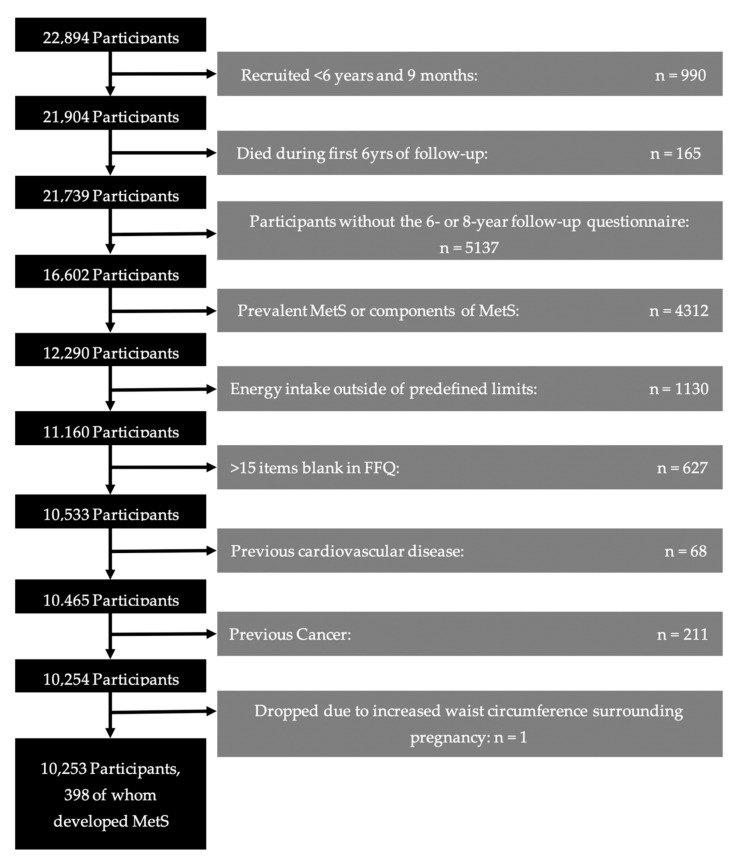
Flow-chart of participants in the SUN Project.

**Table 1 antioxidants-12-00686-t001:** Baseline characteristics of the SUN Participants according to baseline total coffee consumption.

	Coffee Consumption			
	<1 Cup/Month	≥1 Cup/Month–<1 Cups/Day	≥1 Cup/Day–<4 Cups/Day	≥4 Cups/Day
Age (y)	33 (10)	34 (11)	37 (10)	39 (10)
Sex (% women)	65.9	65.5	68.5	65.2
Energy intake (kcal/d)	2333 (605)	2344 (603)	2404 (571)	2469 (648)
Adherence to MedDiet (0–8 score)	3.8 (1.8)	3.8 (1.7)	4.0 (1.7)	4.0 (1.6)
Body-mass index (kg/m^2^)	22.3 (2.6)	22.6 (2.7)	22.7 (2.7)	23.2 (2.9)
Physical activity (METs–h/wk)	24.4 (27.7)	23.5 (24.2)	20.5 (20.7)	22.5 (25.2)
Smoking (%)				
Current smokers	14.8	18.7	24.7	39.3
Former smokers	17.4	21.2	28.9	28.2
Pack-years of smoking	2.6 (5.9)	3.2 (6.6)	5.3 (7.9)	9.1 (10.7)
Prevalent depression (%)	8.8	9.6	10.2	11.3
Years at university (yrs)	4.9 (1.5)	4.9 (1.4)	5.1 (1.5)	5.1 (1.6)
TV watching (h/day)	1.6 (1.2)	1.6 (1.2)	1.6 (1.2)	1.6 (1.2)
Alcohol intake (g/day)	4.2 (7.2)	5.0 (6.8)	6.5 (8.9)	6.6 (10.2)
Siesta (%)				
≤half an hour	31.6	32.0	36.4	34.3
>half an hour	35.3	38.4	30.3	34.5
Adding sugar to beverages (%)	19.2	28.0	29.7	30.5
Family history of diabetes (%)	10.7	12.7	14.6	17.4

Numbers in the table show the mean (SD), unless otherwise stated. Yrs, years; d, day; MET, metabolic equivalent tasks.

**Table 2 antioxidants-12-00686-t002:** Odds Ratio (and confidence intervals) for metabolic syndrome according to total coffee consumption.

	Coffee Consumption					
	<1 Cup/Month	≥1 Cup/Month–<1 Cups/Day	≥1 Cup/Day–<4 Cups/Day	≥4 Cups/Day	*P* for Trend	Per +1 Cup/Day
Cases/N	48/1312	82/2184	247/5983	21/376		
Crude Model	1 (ref)	1.03 (0.71–1.47)	1.13 (0.82–1.55)	1.53 (0.90–2.58)	0.115	1.07 (0.99–1.15)
Age- and sex-adjusted model	1 (ref)	0.92 (0.63–1.34)	0.88 (0.64–1.22)	1.04 (0.61–1.79)	0.833	1.00 (0.93–1.09)
Multivariable Adjusted Model	1 (ref)	0.79 (0.53–1.16)	0.71 (0.50–0.99)	0.73 (0.42–1.29)	0.147	0.94 (0.87–1.03)

1: adjusted for sex, age, years at university, hours of watching TV (continuous), smoking (three categories), pack-years of smoking (continuous), body mass index (continuous), physical activity (quartiles), total energy intake (continuous), adherence to the Mediterranean diet (continuous), added sugar to beverages, prevalent depression, siesta (three categories), alcohol intake (three categories), and family history of diabetes. Abbreviations: N = total number within the category. Results were obtained from logistic regression models. *p* values < 0.05 were deemed statistically significant.

**Table 3 antioxidants-12-00686-t003:** Odds Ratio (and confidence intervals) for metabolic syndrome according to caffeinated coffee consumption.

	Caffeinated Coffee Consumption				
	<1 Cup/Month	≥1 Cup/Month–<1 Cups/Day	≥1 Cup/Day–<4 Cups/Day	≥4 Cups/Day	*P* for Trend
Cases/N	90/2290	65/1904	226/5342	17/319	
Crude Model	1 (ref)	0.87 (0.63–1.20)	1.08 (0.84–1.38)	1.36 (0.80–2.31)	0.119
Age- and sex-adjusted model	1 (ref)	0.81 (0.58–1.14)	0.92 (0.71–1.19)	1.00 (0.58–1.73)	0.903
Multivariable Adjusted Model	1 (ref)	0.73 (0.53–1.04)	0.78 (0.59–1.02)	0.75 (0.43–1.33)	0.274

1: adjusted for sex, age, years at university, hours of watching TV (continuous), smoking (three categories), pack-years of smoking (continuous), body-mass index (continuous), physical activity (quartiles), total energy intake (continuous), adherence to the Mediterranean diet (continuous), added sugar to beverages, prevalent depression, siesta (three categories), alcohol intake (three categories), a family history of diabetes, and decaffeinated coffee consumption (three categories). Abbreviations: N = total number within the category. Results were obtained from logistic regression models. *p* values < 0.05 were deemed as statistically significant.

**Table 4 antioxidants-12-00686-t004:** Odds Ratio (and confidence intervals) for metabolic syndrome according to decaffeinated coffee consumption.

	Decaffeinated Coffee Consumption			
	<1 Cup/Month	≥1 Cup/Month–<1 Cups/Day	≥1 Cup/Day	*P* for Trend
Cases/N	266/6144	85/2585	47/1126	
Crude Model	1 (ref)	0.76 (0.59–0.97)	0.96 (0.70–1.32)	0.844
Age- and sex-adjusted model	1 (ref)	0.85 (0.66–1.10)	0.90 (0.65–1.25)	0.563
Multivariable Adjusted Model	1 (ref)	0.90 (0.69–1.17)	0.79 (0.56–1.11)	0.182

1: adjusted for sex, age, years at university, hours of watching TV (continuous), smoking (three categories), pack-years of smoking (continuous), body mass index (continuous), physical activity (quartiles), total energy intake (continuous), adherence to the Mediterranean diet (continuous), added sugar to beverages, prevalent depression, siesta (three categories), alcohol intake (three categories), a family history of diabetes, and caffeinated coffee consumption (four categories). Abbreviations: N = total number within the category. Results were obtained from logistic regression models. *p* values < 0.05 were deemed as statistically significant.

## Data Availability

Available upon request from the Department of Preventive Medicine and Public Health, School of Medicine, University of Navarra, Pamplona, Spain.

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
