# Peer review of "Coffee Consumption and the Risk of Metabolic Syndrome in the ‘Seguimiento Universidad de Navarra’ Project"

_antioxidants, 2023, doi:10.3390/antiox12030686_

Round 1

Reviewer 1 Report

The evidence on the relationship between coffee consumption and the risk of metabolic syndrome (MetS) has been discussed.  The research group have therefor tried to investigate the association between coffee consumption and incident MetS by using the ‘Seguimiento de Navarra’ cohort SUN project, which include 10,253 participants. Coffee consumption was assessed at baseline using a previously validated food-frequency questionnaire and the development of MetS was collected after 6 years of follow-up. In this cohort there seems that moderate coffee consumption may be associated with a lower risk of MetS. Some clear limitations affect the study as it  was self-reported rather than quantified by objective means.

Comments:

1)     We know that coffee contains a lot of bioactive components. We also know that is very important to include method of preparation for the coffee; Filter coffee, France coffee or expresso, as for filter coffee and instant coffee a lot of the bioactive compounds are filtered out.  Have the authors any information that could support their data re. coffee  preparation brewing methods?

2)      None of the data shown in the study are significant. Please explain the theory behind the Hosmer-Lemeshow test.

3)     why didn't the authors do a meta-analysis instead?

Author Response

Reviewer 1

The evidence on the relationship between coffee consumption and the risk of metabolic

syndrome (MetS) has been discussed.  The research group have therefor tried to investigate

the association between coffee consumption and incident MetS by using the ‘Seguimiento de Navarra’ cohort SUN project, which include 10,253 participants. Coffee consumption was

assessed at baseline using a previously validated food-frequency questionnaire and the

development of MetS was collected after 6 years of follow-up. In this cohort there seems that

moderate coffee consumption may be associated with a lower risk of MetS. Some

clear limitations affect the study as it  was self-reported rather than quantified by objective

means.

We thank the reviewer for the thorough and pertinent revision of the manuscript.

  1. We know that coffee contains a lot of bioactive components. We also know that is very important to include method of preparation for the coffee; Filter coffee, France coffee or expresso, as for filter coffee and instant coffee a lot of the bioactive compounds are filtered out.  Have the authors any information that could support their data re. Coffee preparation brewing methods?

We agree with the reviewer that this is an important issue. As we already stated in the discussion section, espresso and percolator coffee are the predominant brewing methods in Spain according to previous literature (Eur J Nutr 2019,58:1415–1427). Unfortunately, as we acknowledge in the limitation section, we have no available information on the brewing method our participants used.

  1. None of the data shown in the study are significant. Please explain the theory behind the Hosmer-Lemeshow test.

In table 2, there is a statistically significant result for the reduction in MetS in participants who consumed ≥1 cup/day to <4 cups/day after using a multivariable adjusted model.

The Hosmer-Lemeshow test addresses the goodness of fit of the logistic model and, as such, the calibration of the model. We have included more detailed information on this in the methods section as we now provide a reference for this method.

  1. Why didn’t authors use a Metanalysis instead?

Although a metanalysis is the highest form of scientific evidence, we believe that conducting a meta-analysis goes beyond the scope of the present manuscript. In this manuscript, we provide primary data from a large prospective study. Nevertheless, we will consider this suggestion for a future publication.

Reviewer 2 Report

The manuscript is concerned with the evaluation of the relationship between coffee consumption and the risk of metabolic syndrome. The study was conducted with a large number of participants.

However, in the text (line 83) there is a different number of people included in the study than in the figure?

In the Statistical analysis (2.5) should be rather logistic regression than logistical.

Since the study was conducted on such a large group, it would be useful to divide the respondents depending on the method of brewing coffee or whether, for example, coffee was drunk with milk or not. It is known that factors such as the method of brewing or the addition of milk can significantly affect the nutritional value of coffee, which can affect the obtained results.

The list of references should be adapted to the requirements of the journal, the style of writing page numbering of articles and the style of writing article titles should be unified.

Author Response

Reviewer 2

The manuscript is concerned with the evaluation of the relationship between coffee

consumption and the risk of metabolic syndrome. The study was conducted with a large number of participants.

We appreciate this comment and thank the reviewer for the comments which we believe helped us improve our manuscript.

  1. However, in the text (line 83) there is a different number of people included in the study than in the figure?

Thank you. This error has been remedied.

  1. In the Statistical analysis (2.5) should be rather logistic regression than logistical.

Also corrected accordingly. Thank you.

  1. Since the study was conducted on such a large group, it would be useful to divide the respondents depending on the method of brewing coffee or whether, for example, coffee was drunk with milk or not. It is known that factors such as the method of brewing or the addition of milk can significantly affect the nutritional value of coffee, which can affect the obtained results.

Indeed, it would be interesting to see stratified results according to brewing method and according to addition of milk to coffee. However, variables such as adding milk and brewing method were not available in the questionnaire. We agree that these are very important factors and could be partially responsible for the lack of definitive results.

  1. The list of references should be adapted to the requirements of the journal, the style of writing page numbering of articles and the style of writing article titles should be unified

Thank you. We have revised the references accordingly.

Reviewer 3 Report

An interesting topic to cover especially that there is increased interest in exploring dietary approaches to manage and even prevent the metabolic syndrome worldwide. This manuscript can be considered for publication but only after addressing the comments below. Also, English and format editing is needed in different parts of the manuscript.

. Abstract: to support the rationale for this study, authors should also refer to the increased prevalence of metabolic syndrome worldwide and the importance of prevention and management via dietary approaches including coffee consumption in the Background section. In the Methods section line 26, it is not clear what self-reported means; also, how was MetS defined given the different existing definitions? Authors need to use a more specific language specially when describing their methodology: what are these 4 categories for categorizing coffee consumption? In the Results section, authors should report findings when adjusting for covariates. 

. Introduction: important background information including recent studies and meta-analysis studies on coffee consumption and the metabolic syndrome is missing, such as a recent meta-analysis by Ramli et al, 2021. Also, authors should elaborate on the importance of exploring MetS in a Mediterranean context: what are the prevalence figures in Europe including Spain? Also, they should discuss the role that dietary approaches were proven to play in the management and prevention of the metabolic syndrome, and their particular interest in exploring the role of coffee. 

. Materials and Methods: In the section "metabolic syndrome", authors should clarify to readers how participants did self-report those data; what about components such as blood pressure, and blood lipid levels? In the "statistical analysis" section, authors should define what P value is considered significant. 

. Results: Table 1: footnote should be added to explain number in parenthesis and any used abbreviations. Tables 2, 3 and 4: authors should explain any abbreviations used in the table like N. Also, they should remind the readers about the stat analysis used to get the P value and what P value is considered significant. 

. Discussion: when summarizing findings, lines 199-201, authors should clearly state where significant findings were seen versus where trends were seen: this should be clear to the readers. 

. Conclusions: weak and needs to be expanded. What do authors mean by independent prospective studies? When discussing future studies, authors should provide more details and what still needs to be assessed/tested/confirmed in future studies. 

. References: comprehensive review but still missing recent publications in the field of coffee consumption and Mets as indicated in the previous comment. 

Author Response

Reviewer 3

An interesting topic to cover especially that there is increased interest in exploring dietary

approaches to manage and even prevent the metabolic syndrome worldwide. This manuscript can be considered for publication but only after addressing the comments below. Also, English and format editing is needed in different parts of the manuscript.

We thank the reviewer for taking the time to review the manuscript and for the useful comments.

A native speaker has reviewed the manuscript.

ABSTRACT: to support the rationale for this study, authors should also refer to the increased prevalence of metabolic syndrome worldwide and the importance of prevention and management via dietary approaches including coffee consumption in the Background section.

Thank you for your suggestion. We have adapted the background information according to your specifications. We were unable to find clear evidence of global increase in prevalence of metabolic syndrome, but here is the additional information we included in the abstract:

“Metabolic Syndrome (MetS) affects over a third of the United States population with similar prevalence in Europe. Dietary approaches to prevention are important. Coffee consumption has been inversely associated with mortality and chronic disease.”

  1. In the Methods section line 26, it is not clear what self-reported means; also, how was MetS defined given the different existing definitions?

We have added the following information in the abstract to comply with this suggestion:

“All data was self-reported by participants. MetS was defined according to the Harmonizing Definition.”

  1. Authors need to use a more specific language specially when describing their methodology: what are these 4 categories for categorizing coffee consumption?

We have included this information in the current version of the abstract which now reads as follows:

“We used multivariable logistic regression models to estimate odds ratios and 95% confidence intervals for incident MetS according to 4 categories of coffee consumption: <1 cup/month; ≥1 cup/month to <1 cup/day; ≥1 cup/day to <4 cups/day; ≥4 cups/day”

  1. In the Results section, authors should report findings when adjusting for covariates. 

We appreciate this suggestion. We have included in the abstract that the cited odds ratio stemmed from a multivariable-adjusted model.

  1. INTRODUCTION:  important background information including recent studies and meta-analysis studies on coffee consumption and the metabolic syndrome is missing, such as a recent meta-analysis by Ramli et al, 2021. Also, authors should elaborate on the importance of exploring MetS in a Mediterranean context: what are the prevalence figures in Europe including Spain? Also, they should discuss the role that dietary approaches were proven to play in the management and prevention of the metabolic syndrome, and their particular interest in exploring the role of coffee. 

We thank the reviewer for these suggestions. We have included the following sections in the introduction to comply with the reviewer’s suggestions:

“ The most widely accepted definition is the Harmonizing Definition provided by the International Diabetes Federation (IDF) Task Force on Epidemiology and Prevention and the American Heart Association/National Heart, Lung, and Blood Institute (AHA/NHLBI) [1]. Three out of five MetS criteria (which the reader may investigate in the referenced article) must be met in order to make a diagnosis. In the United States, MetS prevalence increased by almost 30% in just over 30 years [2]. European studies have described the prevalence to be 23.9% in men and 24.6% in women [3]. As for Spain, the prevalence of MetS was estimated to be 22.7%, according to 2008-2010 data and using the Harmonized Definition [4]. This high prevalence is troubling as the presence of MetS is associated with increased risk of new onset cardiovascular disease (CVD) and all-cause mortality [5]. The presence of MetS doubles the risk of CVD over 5-10 years and increases the risk of type 2 diabetes fivefold or more over a lifetime [1]. It is therefore of clinical relevance to work towards its prevention [1].

Diet and lifestyle changes are an important tool in the prevention of chronic disease. The Mediterranean diet is a pattern of food consumption that has proven its potential to prevent cardiovascular health deterioration [6] as well as MetS [7]. It differs from other dietary patterns because of its comparatively higher consumption of fats (mainly olive oil) and moderate wine intake [8].

Coffee is one of the most widely consumed beverages around the world and is consumed at a rate of approximately 8.3kg per year per capita in Europe [9].”

[…]

“A recent metanalysis of randomized controlled trials by Ramli et al. investigated the effects of coffee consumption and green coffee extract intake on anthropometric measurements, glycemic indices, lipid profiles and blood pressure [17] and found all of the these parameters to be improved by green coffee extract supplementation.”

“There are two main varieties of coffee: Arabica and Robusta. The former is more acidic and is predominantly produced in Latin America [23], the latter is stronger but less acidic, mainly found in West Africa and Southeast Asia and used in soluble coffee pro-duction [24]. During the roasting process, the Maillard reaction leads to the oxidative polymerization and degradation of phenolic compounds [25]. This same process reduces the amount of carbohydrates, proteins, chlorogenic acid and free amino acids but increases the lipid and mineral content as well as the caffeine and trigonelline content [26]. The method of coffee preparation has its own role in determining the concentrations of bio-active substances such as diterpenes [27], caffeine or polyphenols [28] For example, Moeenfard et al found that boiled coffee showed the highest diterpene esters concentra-tion, whereas filtered and instant brews showed the lowest concentrations [27].”

  1. MATERIALS AND METHODS: In the section "metabolic syndrome", authors should clarify to readers how participants did self-report those data; what about components such as blood pressure, and blood lipid levels?

We appreciate this suggestion. We have expanded the information on metabolic syndrome ascertainment in the main text of the manuscript. The current version reads as follows:

“Data related to components of metabolic syndrome were assessed on the third and fourth questionnaires, distributed approximately six and eight years after entry into the cohort. Participants self-reported recently measurements of waist circumference, blood pressure, HDL and LDL cholesterols, fasting glucose, as well as diagnoses of hypertension and diabetes and any medicines they were using. MetS was defined according to the Harmonizing Definition from the IDF and AHA/NHLBI. That is, it was defined as the presence of 3 of the following 5 criteria [1]: (a) Central adiposity (waist circumference ≥94 cm for men and ≥80 cm for women, as defined for the European population); (b) elevated triglycerides (≥150 mg/dL or undergoing treatment for hypertriglyceridemia); (c) reduced HDL-cholesterol (<40mg/dL for men and <50 mg/dL for women or undergoing treatment for reduced HDL-cholesterol); (d) high blood pressure (systolic ≥130 mmHg or diastolic ≥85 mmHg or undergoing pharmacological treatment for hypertension); and (e) elevated fasting glucose (≥100 mg/dL or undergoing pharmacological treatment for hyperglyce-mia). Each component of the MetS was addressed as a dichotomous yes/no variable. The resulting ascertainment of MetS has been previously validated in our cohort [33].”

  1. In the "statistical analysis" section, authors should define what P value is considered significant. 

We have included this information in the current version of the manuscript.

  1. RESULTS: Table 1: footnote should be added to explain number in parenthesis and any used abbreviations. Tables 2, 3 and 4: authors should explain any abbreviations used in the table like N. Also, they should remind the readers about the stat analysis used to get the P value and what P value is considered significant

We have changed all the tables accordingly.

  1. DISCUSSION: when summarizing findings, lines 199-201, authors should clearly state where significant findings were seen versus where trends were seen: this should be clear to the readers

Here the following changes were made;

“Our results suggest that moderate coffee consumption (≥1cup/d to <4cups/d) may be associated with a decreased risk of developing MetS in a multivariable model. We did not find any remarkable difference between caffeinated or decaffeinated coffee consumption. Although no single category yielded statistically significant results, there was a trend towards an overall protective effect once covariates are accounted for.”

  1. CONCLUSION: weak and needs to be expanded. What do authors mean by independent prospective studies? When discussing future studies, authors should provide more details and what still needs to be assessed/tested/confirmed in future studies

We appreciate this suggestion. We have expanded the conclusion and changed the wording. We have removed the word independent to avoid misunderstandings. The new version reads:

“In conclusion, moderate and regular coffee consumption might reduce the risk of developing MetS, but further evidence is needed. There are physiological mechanisms that support a relationship between coffee and MetS. However, currently available cohort studies are scarce. Out study is of the first to find some statistically significant associations though further prospective studies are warranted to confirm these results.”

  1. REFERENCES: comprehensive review but still missing recent publications in the field of coffee consumption and Mets as indicated in the previous comment.

We have added the aforementioned study as well as several other when expanding upon the background in our introduction.

Round 2

Reviewer 2 Report

The manuscript has been revised and may be published in its current form.

Reviewer 3 Report

Authors have addressed all comments.